

# Seamless seasonal to multi-annual predictions of temperature and standardized precipitation index by constraining transient climate model simulations

Juan C. Acosta Navarro[1], Alvise Aranyossy[2], Paolo De Luca[2], Markus G. Donat[2,3], Arthur Hrast
Essenfelder[1], Rashed Mahmood[4], Andrea Toreti[1] and Danila Volpi[1]

[1]European Commission, Joint Research Centre, Ispra, Italy

[2]Barcelona Supercomputing Center (BSC), Barcelona, Spain

[3]Institució Catalana de Recerca i Estudis Avançats (ICREA), Barcelona, Spain

[4]National Center for Climate Research (NCKF), Danish Meteorological Institute, Copenhagen, Denmark

*Correspondence to*: Juan C. Acosta Navarro (juan.acosta-navarro@ec.europa.eu)

**Abstract.** Seamless climate predictions integrate forecasts across various timescales to provide actionable information in sectors such as agriculture, energy, and public health. While significant progress has been made, there is still a gap in the continuous provision of operational forecasts, particularly from seasonal to multi-annual time scales. We demonstrate that filling this gap is possible using an established climate model analog method to constrain variability in CMIP6 climate simulations. The analog method yields predictive skill for surface air temperature forecasts across timescales, ranging from seasons to several years, consistently outperforming the unconstrained CMIP6 ensemble. Similar to operational climate prediction systems, standardized precipitation index forecasts are less skillful than surface air temperature forecasts, but still systematically better than the CMIP6 unconstrained simulations. The analog-based seamless prediction system is competitive compared to state-of-the art initialised climate prediction systems that currently provide forecasts for specific time scales, such as seasonal and multi-annual. While the current prediction systems provide only 1-2 initialisations per year, the analog-based system can easily provide seamless predictions with monthly initialisations, delivering seamless climate information throughout the year currently not available from traditional seasonal or decadal prediction systems. Furthermore, due to analog-based predictions being computationally inexpensive, we argue that these methods are a valuable and viable complement to existing operational prediction systems.

## 1 Introduction

Seamless climate prediction aims at integrating and synthesizing climate forecasts over a range of forecast times, from sub-seasonal to multi-decadal time scales (Kirtman et al., 2013; Merryfield et al., 2020; Meehl et al., 2021). It is rooted in the concept that the internal climate variability is not confined to any specific time scale, but instead spans from days to several



decades (Schindler et al., 2015 , Zhang et al., 2020). Seamless climate prediction can support various practical applications, ranging from managing agriculture or water resources, to better preparing for climate-related disasters and in better meeting energy demands (Buontempo et al., 2018, Bett et al., 2022, Sánchez-García et al., 2022).

On subseasonal to seasonal timescales (i.e. from a few weeks to a few months), seamless climate prediction aims to inform about variability associated to phenomena such as the Madden Julian Oscillation (Kim, et al., 2019a), or sudden stratospheric warming events (Sigmond, et al., 2013). Climate and weather variability in these timescales can affect sectors such as agriculture, energy production, and public health (Thomson et al., 2006; Klemm and McPherson, 2017; Kim et al., 2019b; Lledó et al., 2019; Ceglar and Toreti, 2021). Seasonal to multi-annual climate predictions (i.e. from a season to a few years)
provide information to better anticipate climate variations that are externally forced or occur due to natural variability within the climate system, and which include for example the El Niño Southern Oscillation (ENSO, Lopez and Kirtman, 2014), the Indian Ocean Dipole (Shinoda and Han, 2005), or the Arctic (Riddle et al., 2013) and Antarctic Oscillations (Seviour et al. 2014), being important in various sectors including agriculture, water resource management, energy, public health, and disaster risk management (Caron et al., 2015; Solaraju-Murali, et al., 2021; Dunstone et al., 2022). Decadal and multi-
decadal predictions provide information on longer climate trends and variability like the Atlantic Multidecadal Variability (Mann et al., 2014) or Pacific Decadal Oscillation (Liu and Di Lorenzo, 2018) which are essential for long-term planning in infrastructure, resource management, and climate change adaptation (Solaraju-Murali, et al., 2022; Dunstone et al., 2022).

Operational seasonal and decadal climate predictions are produced by integrating forward in time an ensemble of several
parallel climate model simulations forced by a likely external forcing scenario, and initialized from a climate state that is representative of the observed climate (Meehl et al., 2021). The ensemble of model simulations constitutes a pool of equally probable realizations of future climate. After initialization, the models are often subject to shocks followed by a drift away from the observed climate typically towards its own attractor. This can result in a reduction of forecast skill (Bilbao et al., 2021). Model simulations used to deliver climate predictions are computationally expensive, thus being produced only by a
few institutions around the world. Analog-based approaches are alternatives that exploit the freely available large ensembles of non-initialized climate model simulations such as the ones from the Coupled Model Intercomparison Project Phase 6 (CMIP6; Eyring et al., 2016) to produce computationally cheaper climate predictions. These approaches work by scanning for analogs of the observed climate in a large model catalog, typically selecting a subset of them in order to better constrain the variability of those simulations and provide predictability beyond the one determined by the externally forced signal
alone (see methods). An incomplete representation of the climate state at initialization is likely the major disadvantage of the analog-based predictions because of the finite available states present in the multi-model catalog. These states may be less representative or "further away" from the observed target state than the initial states in an initialized climate prediction system. Despite these potential disadvantages from a lack or a more sophisticated initialization, the simulations used in the analog-based predictions are not impacted by initialization shocks or drift, and its direct use is computationally cheap.
Analog-based prediction methods have been successfully applied on seasonal/annual scales by Ding et al., (2018) and (2019)





to predict climate in the tropics (e.g. multi-year ENSO forecasts), and by Mahmood et al., (2021) and (2022), De Luca et al. (2023) and Donat et al. (2024) on decadal to multi-decadal timescales to give an outlook beyond the available operational decadal predictions.

Despite recent progress in existing prediction systems (Kirtman et al., 2013; Merryfield et al., 2020; Meehl et al., 2021), only operational decadal predictions provide information across these different time scales, yet this information is only available typically at the beginning of each year. Operational seasonal forecasts do provide information about once a month, but the forecast horizon is typically limited to 6 months. For example, a user interested in obtaining climate information for different time scales (e.g. seasonal to multi-annual) would currently have to combine the information from different prediction

systems for the different time scales which are often inconsistent in their set-up, model used, and predictions they provide. In this study we show that this key climate information gap on the seasonal to multi-annual timescales can be filled by exploiting the model analog method to constrain existing non-initialized CMIP6 simulations.

## 2 Methods

We build from the hypothesis that finding the climatic states (analogs) in simulations from a large multi-model ensemble that are closest to an observed target state can provide valuable information on the future evolution of the climate system (Mahmood et al. 2022). The CMIP6 ensemble is currently the largest available pool of simulations from multiple state-of-the-art climate models. In this study we use data from 149 climate simulations from 19 climate models covering the period 1960-2030 forced by historical emissions before 2015 and the SSP2-4.5 scenario emissions afterwards (Table S1).


More specifically, we scan across time and ensemble members for the conditions that better resemble the observed Sea Surface Temperature (SST) anomaly pattern over oceans at a given time as a means to align the natural climate variability around the climatological state of the model to the observed one, which conceptually corresponds to the initialization of climate predictions. To do this, we first estimate the area-averaged, area-weighted ($w$) mean absolute error ($MAE$) of

monthly SSTs for each member of each model ($l$ $in$ $eq.1$), and across years ($k$ $in$ $eq.1$) with respect to the observational reference ($O$ $in$ $eq.1$) at the desired target month of "initialization" ($m$ $in$ $eq.1$). For example, to produce the Surface Air temperature (TAS) or Standardized Precipitation Index (SPI) forecasts of June-August 2024 with one month lead time, the observed SST anomalies of April 2024 ($m$) are compared with all the April SST anomalies between 1960 and 2030 ($k$) across members in the multi-model ensemble and ranked according to their respective $MAE$, for each member separately. The

modeled SST from all months of April that have the greatest similarity with the target month of April 2024 (i.e. smallest $MAE$) are then selected (analogs of April 2024), and the forecast is constructed by taking the average conditions of the June-August following the selected April analogs. The selection is always done with one month lead time (unless otherwise noted) to provide information well ahead of the targeted forecast period. We found that the analogs generated with an SST pattern comparison for the whole planet is broadly superior to the one using a reduced Indo-Pacific region as in Ding et . (2018)





(Tables S2-S3). Additional sensitivity tests also reveal that the optimal length (m and k) of SST pattern comparison is one
month, independent of the different forecast ranges considered (Tables S2-S7). As opposed to our findings, Mahmood et al
(2022) and Donat et al. (2024) found that for decadal to multi-decadal predictions which are more affected by low-frequency
variability, a constraining based on averages over several years provided the most skillful predictions. The number of
analogs for each TAS (SPI) prediction is defined by the top (top 5) analog(s) in each one of the model simulations which

cover the period 1960 - 2030. Note that for long predictions, the period of analog selection is slightly reduced at both
extremes (e.g. 48 month predictions are based on analogs centered between 1963 and 2027). The number of selected analogs
(i.e. 1 or 5 per member), the number of models and members used, have been determined by performing sensitivity tests
(Tables S8-S13). More specifically, four methods were tested:

- Method 1. All available members from the six models that provide more than 10 members each (Table S1), 122
    members in total.
  - Method 2. Only 10 members from the same six models that provide more than 10 members each, 60 members in
    total.
  - Method 3: Ten members from each model. For models that provide less than 10 members, the members are used

more than once to complete a set of 10 for each model, 190 members in total.
  - Method 4: All available members from the 19 models, 149 members in total.

Using the best overall method from the sensitivity tests (method 4), the selected analogs then constitute the forecasts and can
be interpreted as ensemble members. Additionally, the trend in the TAS analog-based predictions is adjusted by first

removing the signal explained by external forcing as in Smith et al., (2019) and then adding back the externally forced trend
(i.e. the CMIP6 ensemble mean). This step is done because the observed trend is better represented in the CMIP6 ensemble
than in the analog-based predictions without post-processing. For the SPI predictions the trend-adjustment is not needed, as
they are optimal without it.

Mathematically, the criterion to rank and determine the analogs is:

$$MAE_{l,k} = \frac{\sum_i \sum_j w_j |T_{i,j,l,k} - O_{i,j,m}|}{N} \qquad (1)$$

where the indices *i, j, k* and *l* run across longitudes, latitudes, months, and models/members, respectively. *T* stands for the
model values and *N* stands for the total number of ocean grid-points.


We apply the methodology described above to generate retrospective predictions of SPI and TAS of 3 months, 1 year, 2
years and 4 years and evaluate their predictive skill in the period 1962-2018, except for the 3 month predictions which are



only evaluated during 1982-2018, defined by the availability of the benchmark SEAS51 predictions (see next paragraph).
We compute the SPI (McKee et al. 1993) for 3-, 12-, 24- and 48-month accumulations using the R package SPEI (Beguería
and Vicente-Serrano, 2023). Following Smith et al. (2019), the non-forced (or detrended) analog-based predictions and
observations are by definition the residuals that contain only the variability that is not explained by the CMIP6 ensemble
mean. This is achieved by removing through linear regression and at the gridpoint level, the CMIP6 ensemble mean from the
analog-based predictions and observations, respectively. Hence, the non-forced skill throughout the study can be interpreted
as the residual skill explained after removing the externally forced signal. The skill metrics used in this study are the
anomaly correlation coefficient (ACC) and the mean absolute error skill score (MAESS):

$$ACC = \frac{\sum_i^n (F_i - \overline{F})(O_i - \overline{O})}{\sqrt{\sum_i^n (F_i - \overline{F})^2} \sqrt{\sum_i^n (O_i - \overline{O})^2}} \qquad (2)$$

$$MAESS = 1 - \frac{\sum_i^n |(F_i - \overline{F}) - (O_i - \overline{O})|}{\sum_i^n |(R_i - \overline{R}) - (O_i - \overline{O})|} \qquad (3)$$


In eqs. 2 and 3 $F$ stands for forecasted values, $O$ stands for observed values and in eq. 3 the reference $R$ is a trivial
climatological forecast based on observations, the multi-model uninitialized CMIP6 ensemble mean, or the ensemble mean
forecast from an operational prediction system. The letters with the bars denote climatological values. The index $i$ in *eqs. 2
and 3* runs across time.


The list of CMIP6 models used in this study is available in Table S1. All model and observational data have been bilinearly
interpolated to a common grid of 5° X 5° for TAS and about 2.8° X 2.8° for both SSTs (i.e. analog search) and SPI. The
observational reference datasets for SST analog search is ERSSTv5 (Huang et al., 2017), while the observational references
for prediction evaluation of TAS and SPI are based on the monthly averages of Berkeley Earth Surface Temperatures
(BEST, Cowtan, 2023) and the Global Precipitation Climatology Center (GPCC, Becker et al., 2013), respectively. In
addition to the CMIP6 simulations, we used data from two operational climate prediction systems as a benchmark for the
comparisons. For the 3-month predictions, we used the European Centre for Medium-Range Weather Forecasting SEAS5
(Johnson et al., 2019) and for the 12-, 24- and 48-month predictions, we used the initialized climate model EC-Earth3
(Bilbao et al., 2021). Note that both SEAS51 and EC-Earth3 ensembles are limited to 25 members, while the analog
ensemble uses the 149 members from the non-initialized CMIP6 ensemble. Using more members is beneficial, since the skill
of the analog-based predictions clearly increases with ensemble size, regardless of variable or forecast range (Fig. S1).




## 3 Results

### 3.1 Seasonal predictions (3 months)

Figures 1a-f illustrate the spatial distribution of skill for boreal winter (December-February) TAS predictions initialized the 1st of November, as assessed by the ACC and the MAESS. The analog method shows positive statistically significant correlation in the tropics and subtropics, most of the Ocean, and the Arctic (Fig. 1a). A large fraction of skill in these areas can be attributed to the alignment of internal variability in the predictions and observations as revealed by the residual correlation after removing the externally forced signal from CMIP6 (Fig. 1b). In general the analog-based predictions offer added value over a trivial climatological forecast (Fig. 1d) and over the uninitialized CMIP6 ensemble (Fig. 1e), especially over tropical regions. Figure 1c displays the correlation between the observations and SEAS51, an operational seasonal forecasts system, while Figure 1f displays the added value of the analog-based predictions over the SEAS51 ones according to the MAESS. Generally, SEAS51 has higher skill than the analog-based predictions, especially marked in ocean regions like the North Atlantic. However the overall spatial patterns are very similar between the analog-based and SEAS5 predictions, which gives some confidence that both exploit similar sources of predictability. It is worth noting that skill over land of the analog-based predictions is in general statistically not different (p<0.1) between the analog-based and SEAS51 predictions as quantified by MAESS.

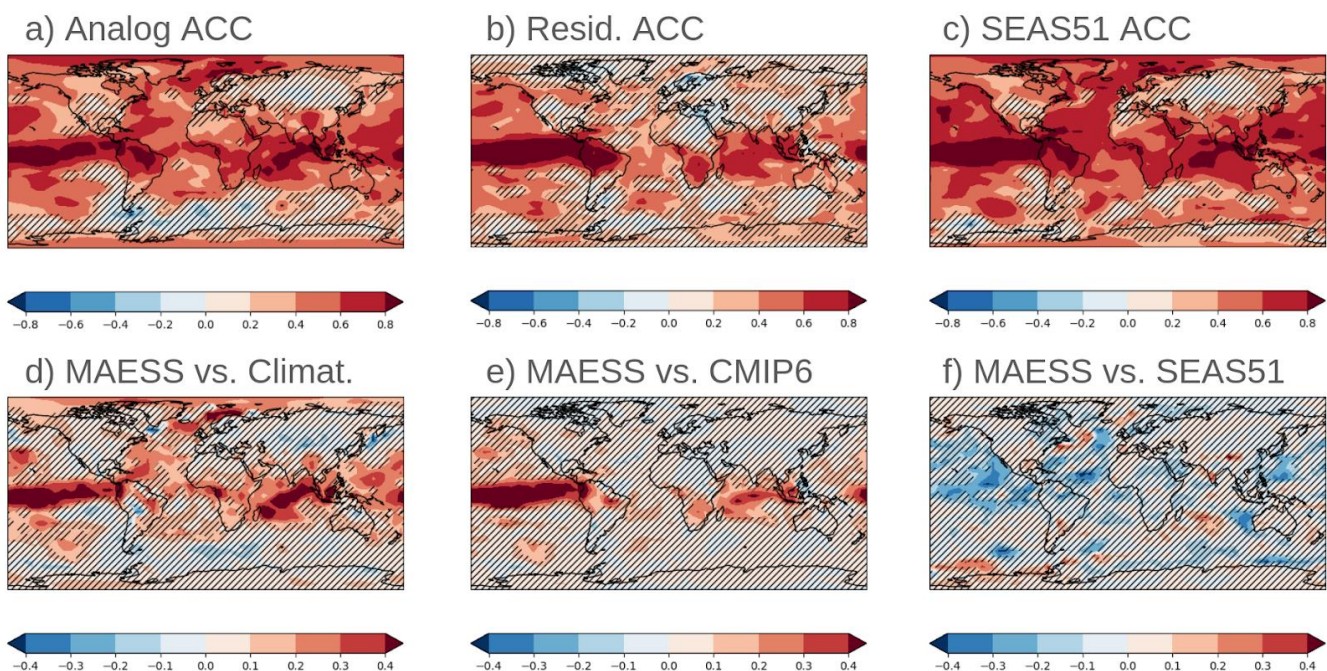

**Figure 1: a) ACC between December-February analog-based ensemble mean predictions and observations of TAS. b) Residual ACC between December-February analog-based ensemble mean predictions and observations of TAS. c) ACC between ECMWF-SEAS51 December-February ensemble mean predictions and observations of TAS. d)**





**MAESS of December-February analog-based ensemble mean predictions of TAS. The reference (R) is a**
**climatological forecast derived from observations. e) MAESS of December-February analog-based ensemble mean predictions of TAS. The reference (R) is the ensemble mean of CMIP6 historical simulations. f) MAESS of December-February analog-based ensemble mean predictions of TAS. The reference (R) are the ECMWF-SEAS51 December-February ensemble mean predictions of TAS. The evaluation period is 1982-2018. The predictions are initialized each November in both analog-based predictions and SEAS51. The hatched regions in all figures indicate statistically**
**non-significant values (p < 0.1) using a two-sided t-test.**

Although not as widespread as December-February predictions, boreal summer (June-August) TAS predictions initialized the 1st of May also display high skill in the tropics. Additionally, skill is also high in many subtropical and mid-latitude regions (Fig. 2a,d). Generally, the skill in northern hemisphere land regions is higher in boreal summer than in winter.
Specifically, the Middle East, Europe and large parts of East Asia show high skill in terms of both ACC and MAESS, although this skill stems primarily from the response to external forcing and not from the analog initialization. Hence, the added value of analog-based predictions over the uninitialized CMIP6 ensemble is mostly limited to tropical and subtropical regions according to the residual correlation (Fig. 2b), but limited to Central America, Southeast Asia and tropical Oceans according to MAESS (Fig. 2e). There is again a very large similarity between the spatial patterns of skill in the analog and
SEAS51 predictions (Fig. 2a,c, respectively). SEAS51 shows larger correlations with observations in general, but similar to boreal winter, the disadvantage of the analog-based predictions over land areas seems limited and mostly not statistically significant with respect to SEAS51 when evaluated with the MAESS (Fig. 2f)





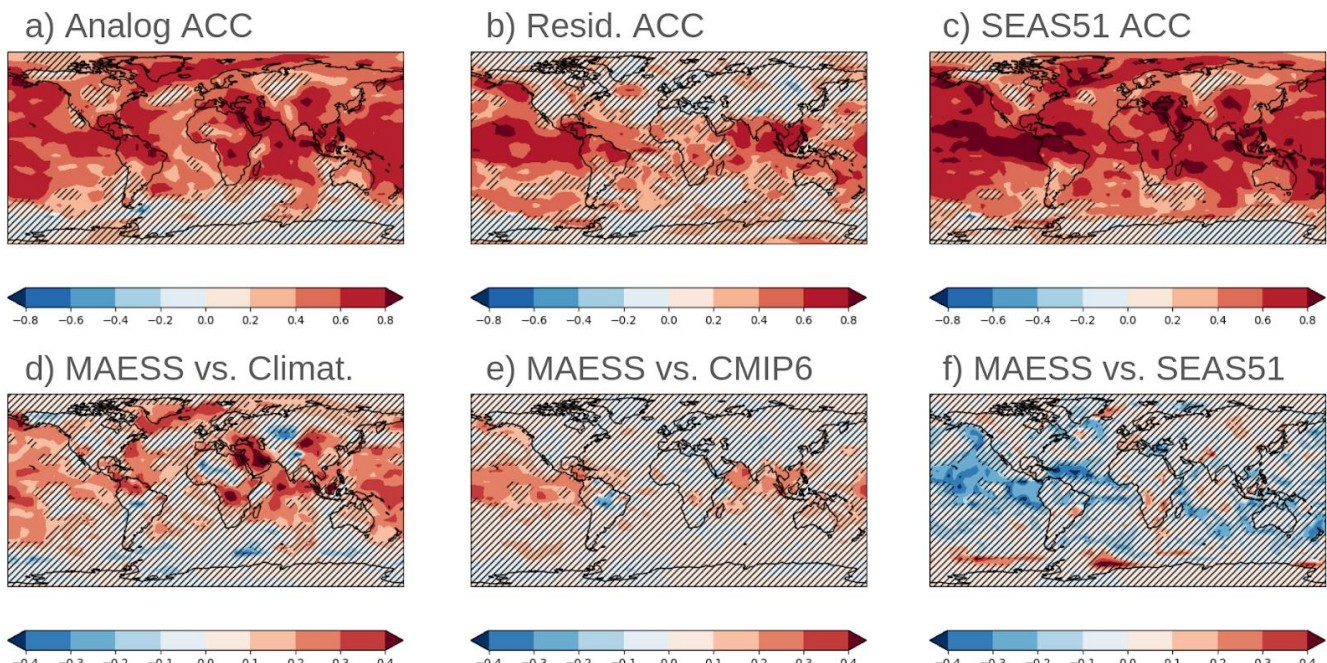

**Figure 2: The same as Figure 1, but for June-August TAS predictions. The predictions are initialized each May.**

Figure 3 shows the correlation coefficients of the SPI3 analog seasonal forecasts and observations during the boreal winter (Fig. 3a-c) and summer (Fig. 3 d-f), respectively. The analog-based predictions exhibit skill in Australasia, southern Africa, and the tropical Americas. In line with what is observed in forecasts from dynamical forecast systems, the analog technique yields predictions for SPI3 that are notably less skillful than those for TAS. Nonetheless, the spatial patterns of regions with skill in the analog and SEAS51 are very similar (Fig. 3a,c,d,f). The residual correlation of precipitation forecasts with the analog method during the boreal winter and summer is shown in Figures 3b,e, respectively. The similarity of these maps to the full skill maps (i.e. Fig. 3a,d), suggests that the analog-based predictions' accuracy is predominantly due to the alignment of natural climate variability in both the models and observations, with one notable exception: the Sahel region in boreal summer (Fig. 3d,e), in which the skill seems to result from the external forcing. It is important to note that the MAESS of both, boreal summer and winter analog-based predictions of SPI3 when using SEAS51 as a reference shows generally non-statistically significant differences, similarly to land TAS (not shown).



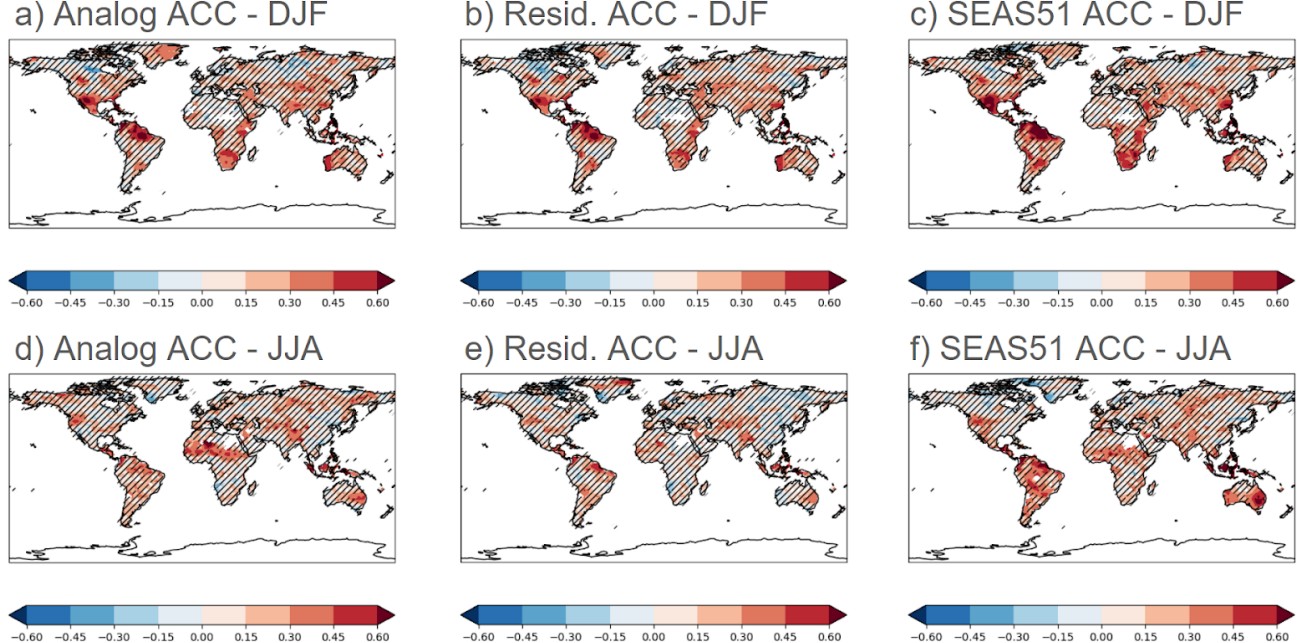

**Figure 3: a) ACC between December-February analog-based ensemble mean predictions and observations of SPI3 b) Residual ACC between December-February analog-based ensemble mean predictions and observations of SPI3 c) ACC between ECMWF-SEAS51 December-February ensemble mean predictions and observations of SPI3. d) ACC between June-August analog-based ensemble mean predictions and observations of SPI3. e) Residual ACC between June-August analog-based ensemble mean predictions and observations of SPI3. f) ACC between ECMWF-SEAS51 June-August ensemble mean predictions and observations of SPI3. The evaluation period is 1982-2018. The predictions are initialized each November (a-c) and each May (d-f). The hatched regions in all figures indicate statistically non-significant values (p < 0.1) using a two-sided t-test.**

Despite the 3-month analog-based predictions being generally less skillful than SEAS51 throughout most of the year, their skill is comparable during boreal fall and winter for both TAS and SPI3 in terms of land area with positive and statistically significant residual correlation (Fig. 4). Although the largest contribution to the skill from the forced signal happens during boreal fall for TAS and SPI3, skill over land peaks around boreal summer and winter for TAS and SPI3, respectively in both analog and SEAS51 predictions. This difference between the two variables can most likely be attributed to a more dominant influence of external forcing on TAS predictability, while for SPI3 the primary driver is natural variability.





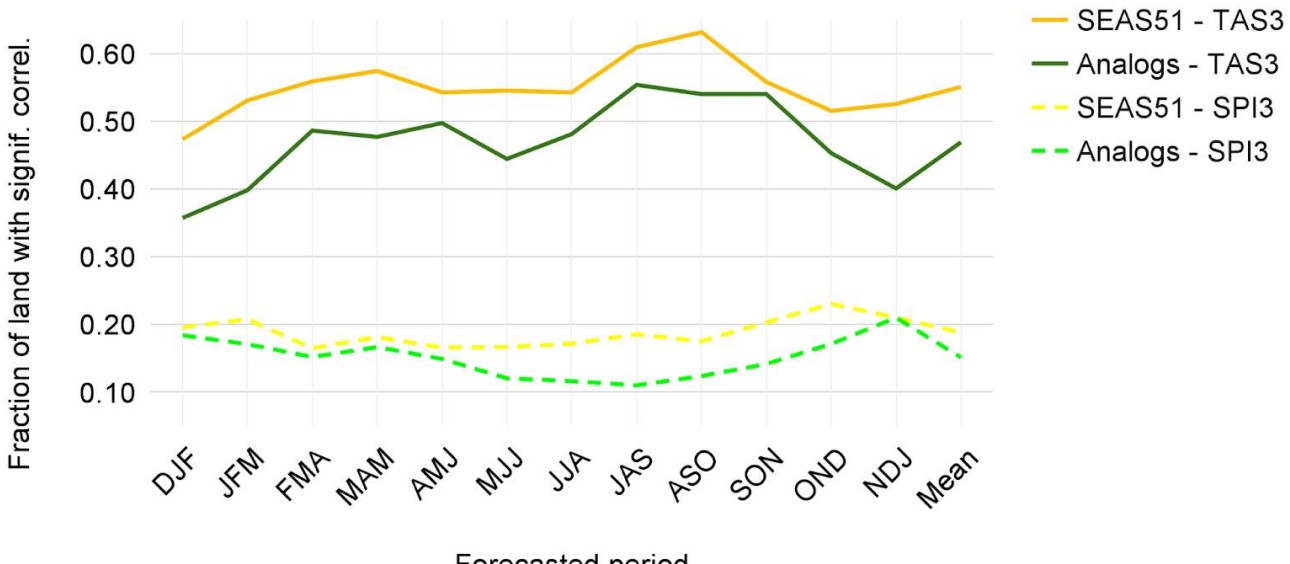

**Figure 4: Fraction of land area with statistically significant positive correlation (p < 0.10) between the 3-month TAS (solid lines) and SPI3 (dashed lines) from the analog, and SEAS51 predictions, and the respective observations using a two-tailed t-test. The evaluation period is 1982-2018. The displayed values are based on residual correlations for**
**TAS to remove the impact from the forced signal.**

Figure 5 shows the temporal evolution of four key seasonal climate indices of the tropical Oceans: The December-February NINO34 (Fang and Xie, 2020) in the tropical Pacific, the June-August tropical Atlantic index (ATL3, Zebiak 1993), and the March-May Western (WIO) and September-November Eastern (EIO, Saji et al., 1999) tropical Indian Ocean indices. All
forecasts are initialized one month before the target season. The indices in the respective seasons are important because they measure oceanic variability that induces remote impacts on hydroclimatic conditions over land. They are a small selection to highlight and confront the analog and the SEAS51 predictions in these particular areas. Although not as skillful as SEAS51, the analog-based predictions of NINO34 and the WIO show high skill and closely follow the observed year-to-year variability and trend. The ATL3 and EIO follow the observed trend but largely underestimate the magnitude of year-to-year
variability as opposed to SEAS51.







**Figure 5: Observed and predicted evolution of the indices estimated as the area-averaged TAS time series of the a) December-February NINO34 (170W-120W, 5S-5N; Fang and Xie, 2020) in the tropical Pacific Ocean, b) the March-May WIO (50E-70E, 10S-10N; Saji et al., 1999) in the tropical western Indian Ocean, c) the June-August ATL3 (20W-0E, 3S-3N; Zebiak 1993) in the tropical Atlantic Ocean, and the September-November EIO (90E-110E, 10S-0N; Saji et al., 1999) in the tropical eastern Indian Ocean.**

## 3.2 Annual and multi-annual predictions (1-4 years)

The skill of annual TAS analog-based predictions is very high (ACC>0.8, MAESS>0.3) across most tropical areas and the North Atlantic, as well as being high, positive and statistically significant over land regions outside of the tropics, as shown in Figure 6a,d. However, skill in areas like central Asia, central North America, and southern South America exhibit mostly low to moderate skill (ACC>0.2), but still surpasses that of a climatological forecast (MAESS>0). The residual correlation of the analog forecasts is illustrated in Figure 6b. The results indicate a distinct improvement of the annual analog forecasts





when compared to the CMIP6 ensemble across the Pacific region, the tropical Atlantic and Indian Oceans, Australasia and East Asia, and large parts of the Americas and Africa. However, based on MAESS, these improvements are limited mostly to the Caribbean, southern Africa and the Maritime Continent (Fig. 6e). This discrepancy between ACC and MAESS is

likely the result of analog-based predictions being capable of estimating the variability around the forced signal (positive ACC), but due to model biases MAESS may be affected to the point of making it zero or negative. There is also a broad similarity in the spatial distribution of skill (correlation) of the analog-based and the operational decadal predictions from EC-Earth3 (Fig. 6a vs. 6c). The analog-based predictions slightly outperform EC-Earth3 over land according to MAESS, as seen in southern Africa or Australia (Fig. 6f)


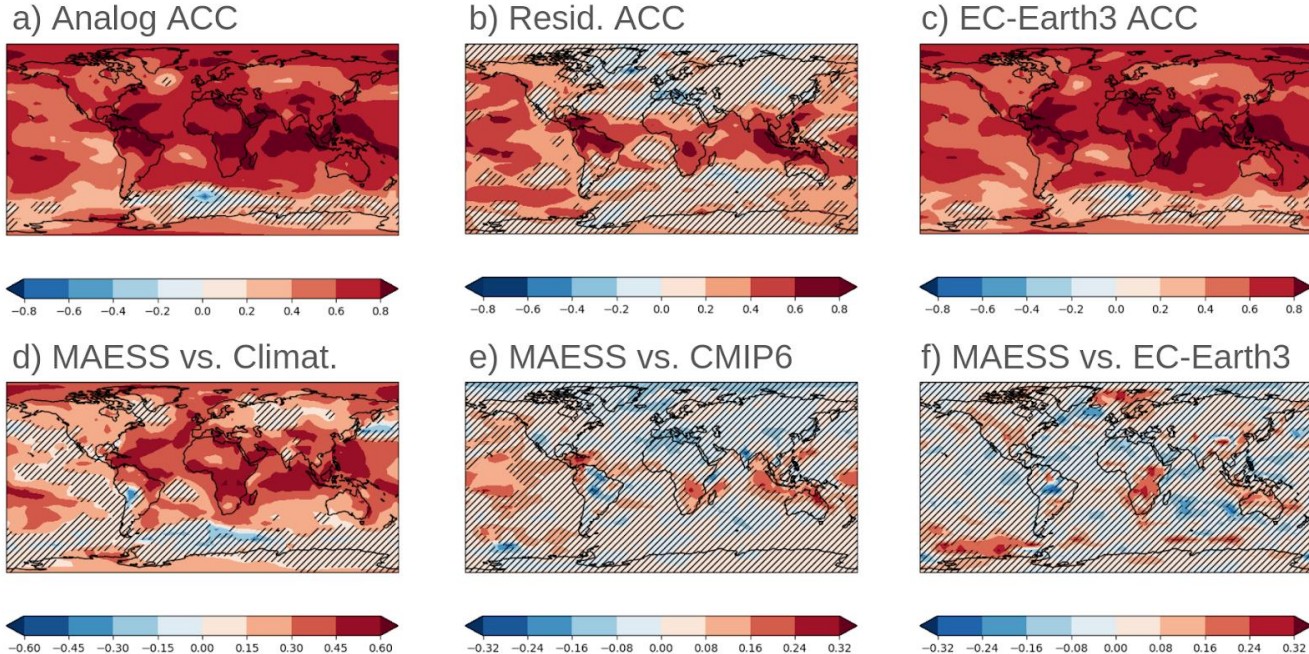

**Figure 6: The same as Figure 1, but for annual TAS predictions (January - December). The evaluation period is 1962-2018. The predictions are initialized each November in both analog-based predictions and EC-Earth3. The dynamic**

**forecasts system evaluated in c,f is EC-Earth3.**

When extending the forecast to a two-year period, the analog-based predictions of TAS continue to show high skill across most land areas. The skill in the extratropical regions such as the Mediterranean or East Asia is comparable with the skill in the tropical zones, as shown by Figures 7a,d. The residual skill after removing the forced signal is slightly less pronounced

for forecasts spanning two years than for one year forecasts, as shown in Figures 7b and 6b, respectively. The benefit from initialization (residual correlation) of the analog-based predictions can still be observed in several areas like tropical South America, South Asia, Australia and Sub-saharan Africa. Most of the Pacific and the Indian Oceans also show benefits from





initialization in the analog-based predictions. Subtropical regions tend to show reduced skill in the TAS analog-based predictions according to the MAESS (Fig. 7e), with the Mediterranean region underperforming CMIP6, but still generally overperforming EC-Earth3 predictions in northern South America, Sub-saharan Africa or Australia (Fig. 7f). As for annual predictions, possible biases present in the analog-based biennial predictions are likely behind the little to no advantage of the analog-based predictions over CMIP6 ensemble based on MAESS (Fig. 7e), despite some clear advantages measured by the residual correlation (Fig. 7b), which only estimates the variability around the forced signal.

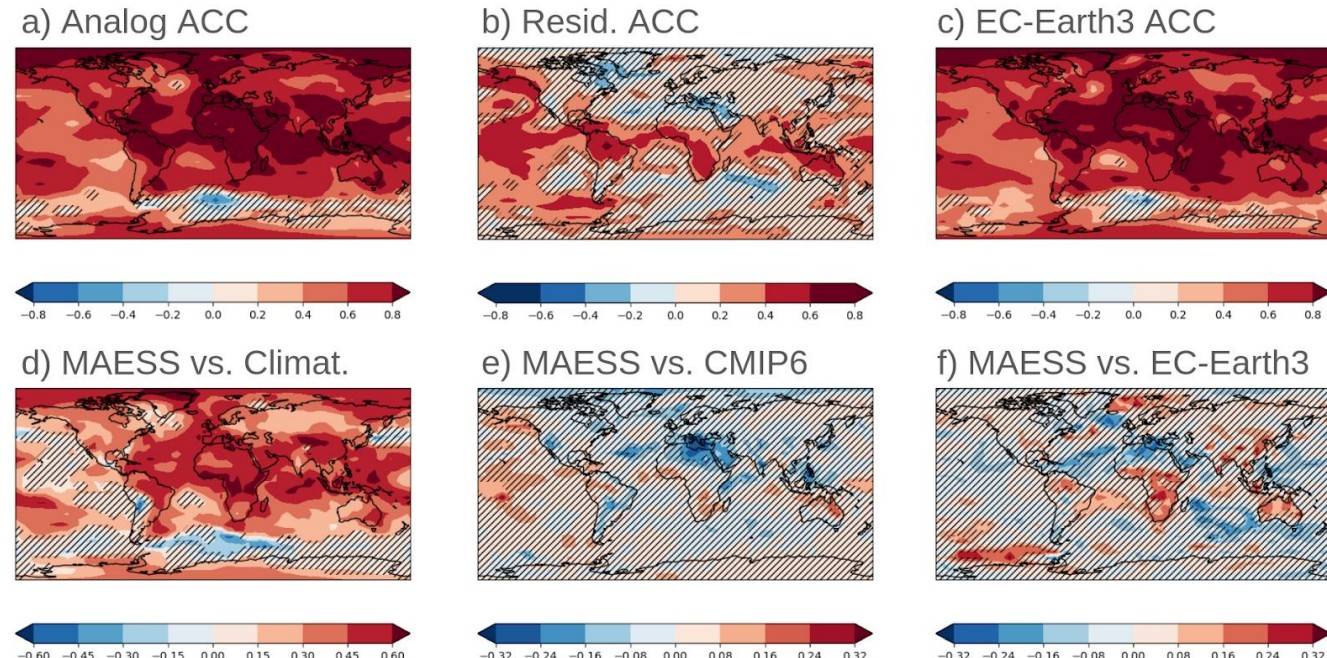

**Figure 7: The same as Figure 1, but for biennial TAS predictions (January - December+1year). The evaluation period is 1962-2018. The predictions are initialized each November in both analog-based predictions and EC-Earth3. The dynamic forecasts system evaluated in c,f is EC-Earth3.**

The results for quadrennial predictions of TAS show higher overall ACC and MAESS than the biennial or annual predictions (Fig. 9 a,c). However, residual correlation is generally smaller in the quadrennial predictions (e.g. the tropical Atlantic and western Africa no longer show added skill). This implies that despite higher overall ACC, the benefit from initialization is smaller for quadrennial predictions than for biennial and annual. Furthermore, the analog-predictions clearly underperform CMIP6 when measured by MAESS (Fig. 8e) as opposed to the overall clear advantage measured by residual ACC (Fig. 8b). This is especially clear in the northern hemisphere subtropics and again most likely the result of model biases. The added value of the analog prediction over EC-Earth3 is however still visible in many regions, with the analog-based predictions underperforming EC-Earth3 predictions only in the North Atlantic, northern Africa and southwestern Asia, but





overperforming it in northern South America, Sub-saharan Africa, South and Southeast Asia, Australia and parts or Europe

(Fig. 8f).

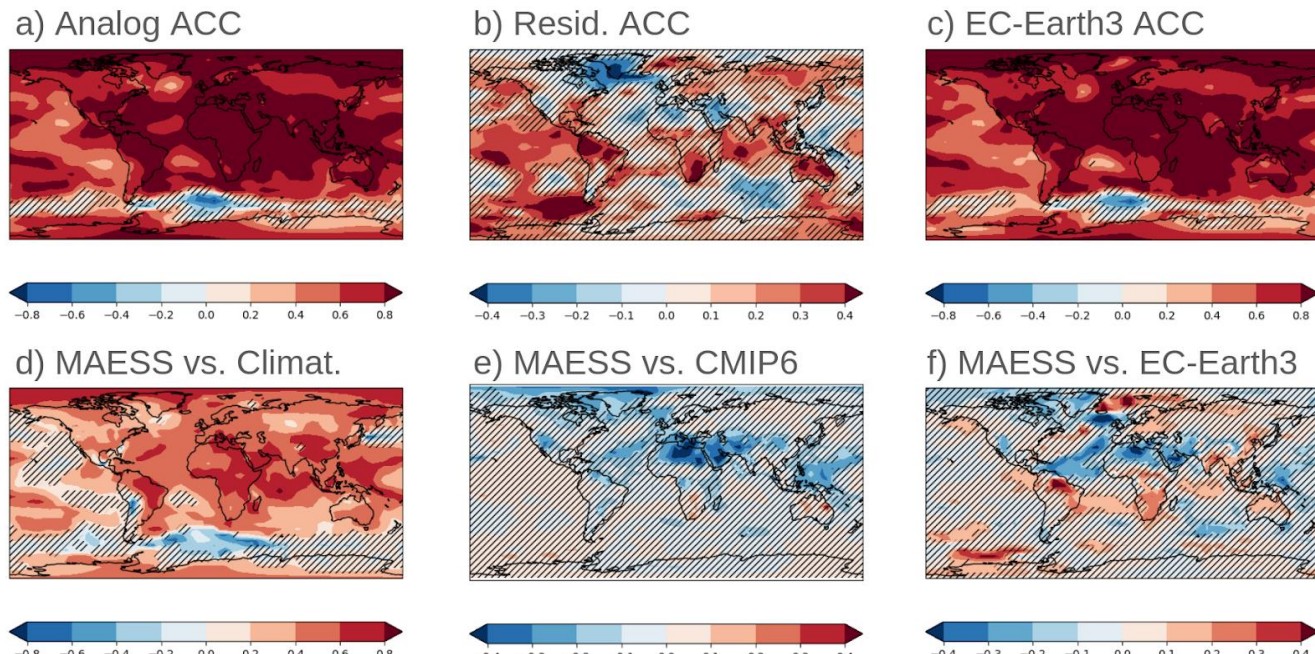

**Figure 8: The same as Figure 1, but for quadrennial TAS predictions (January - December + 3 years). The evaluation period is 1962-2018. The predictions are initialized each November in both analog-based predictions and EC-Earth3.**

**The dynamic forecasts system evaluated in c,f is EC-Earth3.**



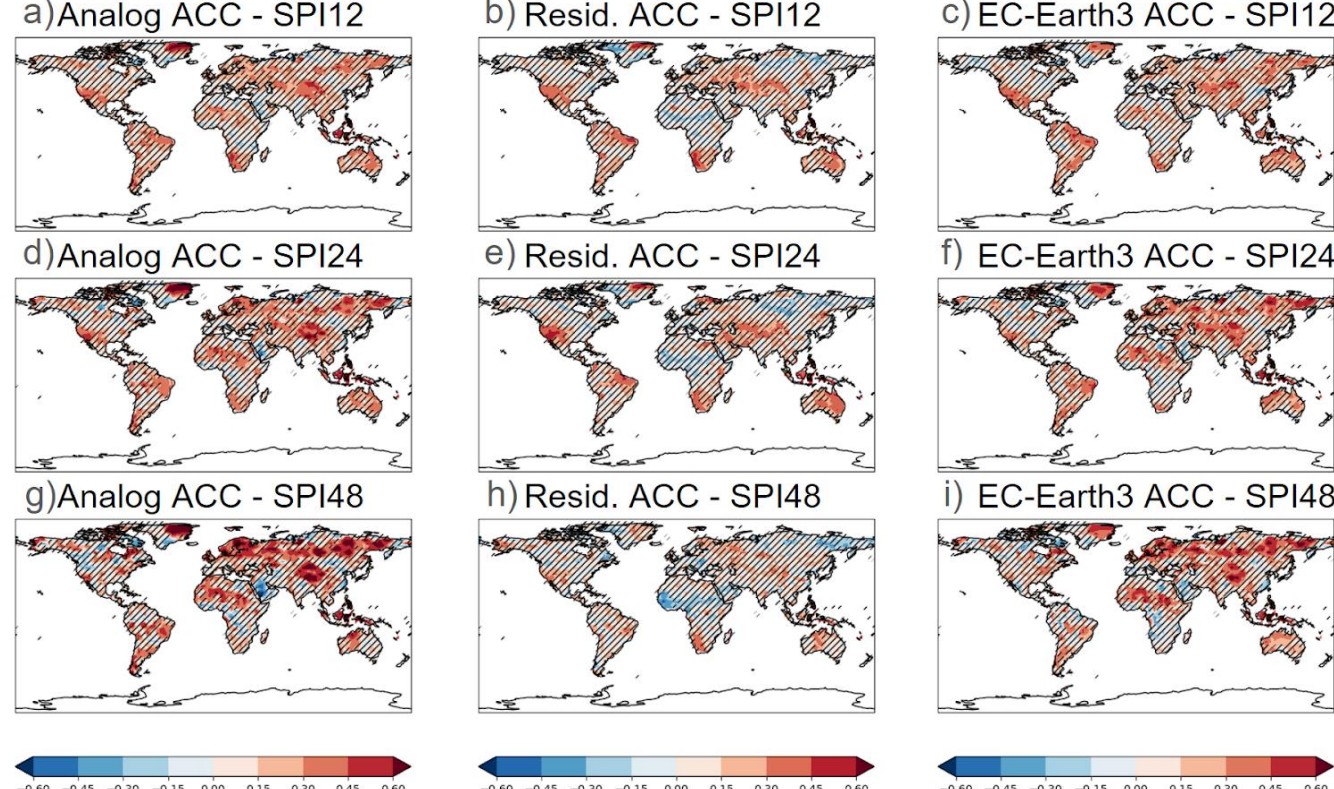

**Figure 9: The same as Figure 3a-c, but for a-c) SPI12, d-f) SPI24 and g-f) SPI48 predictions initialized every November. The reference period is 1962-2018. The dynamic forecast system evaluated in c,f,i is EC-Earth3.**


The correlation maps of SPI12 (Fig. 9a,c), SPI24 (Fig. 9d,f) and SPI48 (Fig. 9g,i) predictions using the analog method and EC-Earth3 are again spatially similar. There is an increase of correlation in northern hemisphere high latitudes with longer precipitation accumulations (SPI48), but comparable skill elsewhere in SPI12, 24 and 48, except for a few regions such as southern Africa or western North America which exhibit lower skill at longer accumulations. An important fraction of the skill for SPI24 and especially SPI12 predictions stems from the synchronization of unforced variability in the models and observations, similar to the seasonal predictions. This can be implied by the broad similarities between Figures 9a,d and 9b,e. Contrastingly, for SPI48 predictions, the forced signal largely dominates over the non-forced one. MAESS maps of SPI12, 24 and 48 reveal mostly non significant values when comparing analog-based predictions with both CMIP6 and EC-Earth3 (not shown).





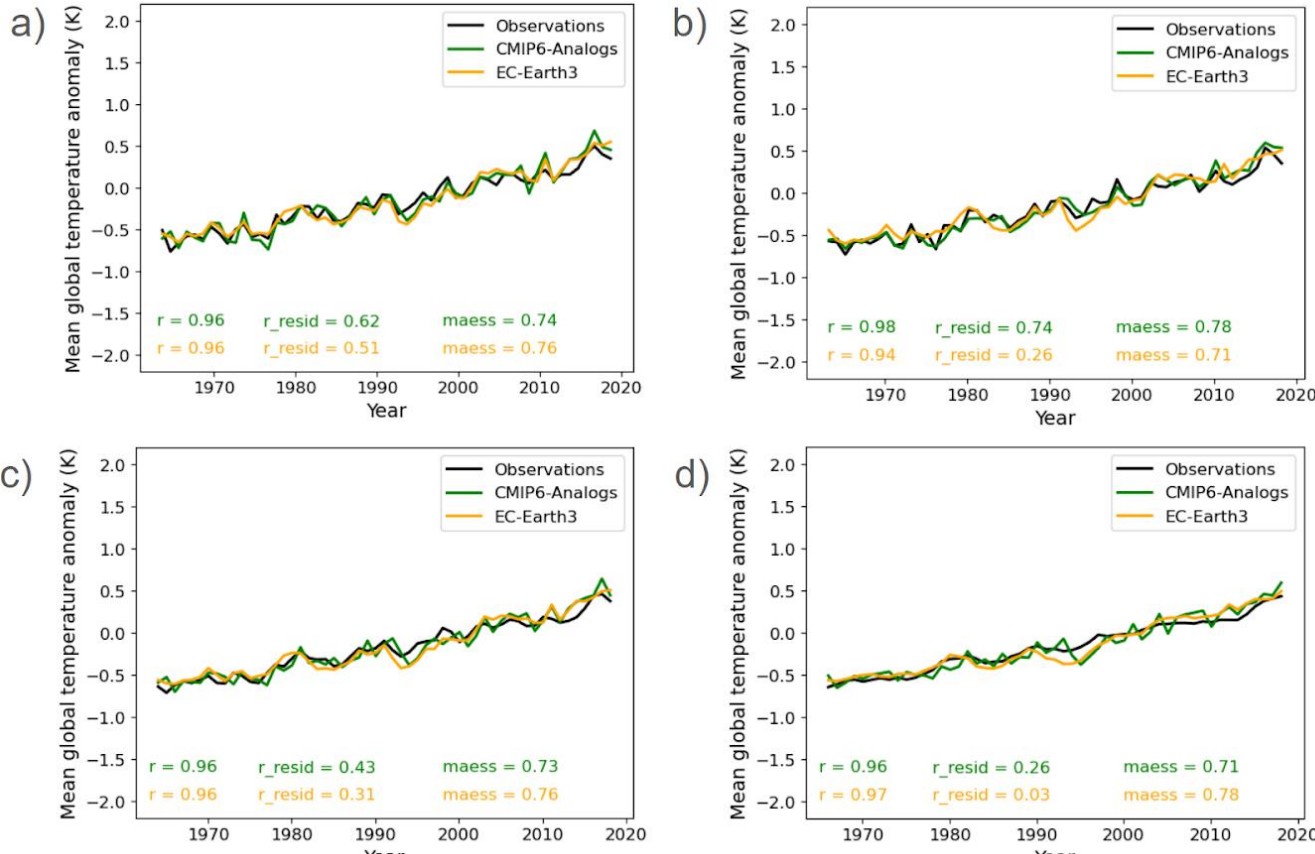

**Figure 10: Observed and predicted evolution of the global area-averaged TAS time series of a) annual January-December predictions initialized in November for both analog-based method and EC-Earth3 (2-month lead), b) the annual July-June(+1 year) predictions initialized in November (8-month lead) for EC-Earth3 and in June (1-month lead) for the analog-based method, c) the biannual January-December(+1 year) predictions initialized in November for both analog-based method and EC-Earth3 (2-month lead), and d) the quadrennial January-December(+3 years) predictions initialized in November for both analog-based and EC-Earth3 (2-month lead).**

Figure 10 presents the time series for annual (a,b), biennial (c), and quadrennial (d) averages of global TAS in the analog-based predictions, EC-Earth3 predictions and the observations. Annual predictions initialized each November for the following January-December (Fig. 10a) show very similar results with comparable performance metrics for the analog and EC-Earth3 predictions. When considering the 12-month period from July-June (Fig. 10b), the analog-based predictions initialized in June (1-month lag) are superior to the EC-Earth initialized the previous November (8-month lag), showing residual correlations of 0.74 and 0.26 for the analog-based and EC-Earth3 predictions, respectively. This example highlights a key advantage of the analog-based predictions over the dynamical prediction systems, as the analog ones can be produced every month throughout the year without large computational cost as opposed to the dynamical ones. Biennial and




quadrennial predictions (Fig. 10c,d) initialized each November (one month lead), show very similar values of correlation for both analog and EC-Earth3 predictions, with residual correlations higher in analog-based predictions and MAESS slightly higher in EC-Earth3 predictions.


Figure 11 summarizes the results of the analog and EC-Earth3 predictions in terms of total land area fraction with significant positive correlation for annual, biennial and quadrennial predictions of TAS and SPI. Due to a saturation of correlation with observations, the residual correlation after removing CMIP6 signal is used for TAS, while the anomaly correlation of the actual time series is used for SPI. The analog and EC-Earth3 predictions initialized every November have a comparable skill

of predicting 12-month TAS and SPI (Fig. 11 a,d) that decreases with increasing lead time from 2 months up to 13 months ( dark green and yellow lines). However, when using analog-based predictions every month always with one month lead-time, the skill is broadly superior (light green line). The skill increases when the forecasts are initialized after the spring ENSO predictability barrier. For 24 month forecasts, the November initialized analog-based predictions are slightly superior to EC-Earth for both variables (Fig. 11b,e) and the analog-based predictions initialized every month are consistently superior to the

November initialized ones, similar to the 12 month forecasts. Analog-based predictions of 48-month TAS and SPI initialized every November are less skillful than the EC-Earth3 counterpart, while the analog-based predictions initialized every month have on average a similar skill than EC-Earth3, but exhibit more variability throughout the year depending on the month of initialization and variable predicted (Fig. 11 c,f).

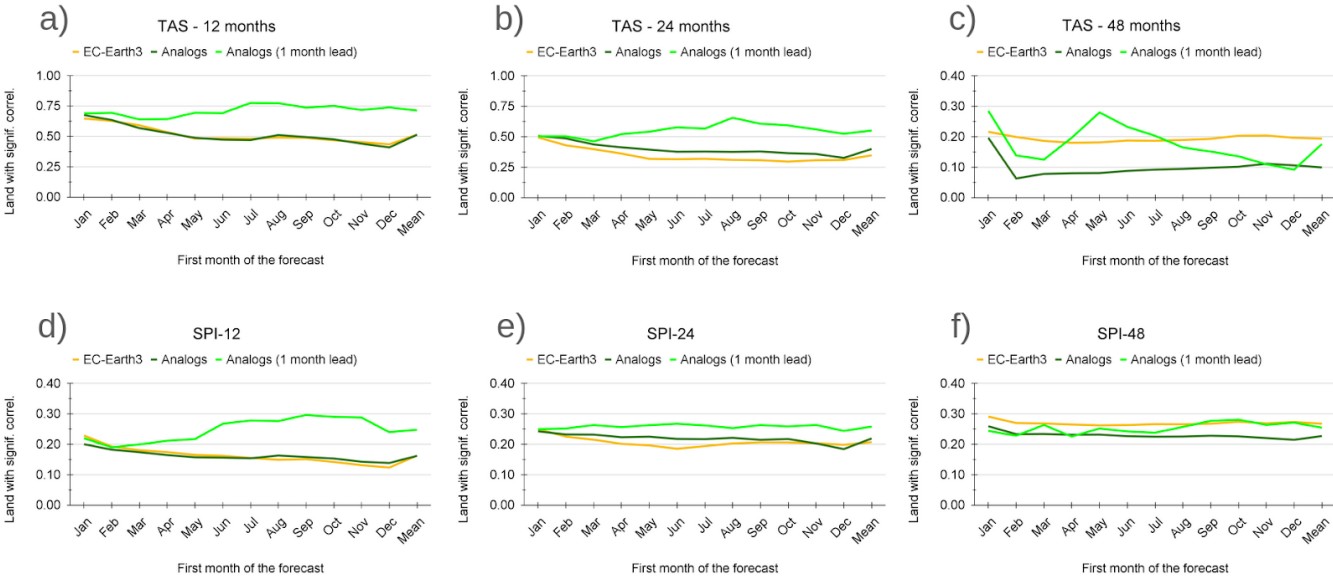


**Figure 11: Fraction of global land area with statistically significant (p<0.1) positive residual correlation between TAS predictions and observations for a) 12-month, b) 24-month and c) 48-month forecasts. The panels d), e) and f) are the same as panels a), b) and c), respectively, but for the statistically significant (p<0.1) positive correlation between SPI predictions and observations using a two-sided t-test. The dark green and yellow lines in all panels show the skill of**





**the analog-based and EC-Earth3 predictions, respectively, initialized every November as the lead-time increases from 2 to 13 months. The light green line shows the skill of the analog-based predictions, initialized always with a one-month lead-time.**

## 4. Summary and conclusions

The analog-based predictions demonstrate skill on seasonal to multi-annual time scales, in many cases comparable to state-of-the-art numerical prediction systems developed for either seasonal or decadal climate predictions. On seasonal timescales the analog-based predictions demonstrate high skill for boreal winter and summer TAS forecasts with one-month lead time, particularly in the tropics, North Atlantic, and most of the Arctic, offering substantial added value over the CMIP6 ensemble (e.g. residual skill). As for boreal summer, the skill extends into subtropical and mid-latitude regions, with the northern
hemisphere land showing greater skill in summer than winter. The improvement over the non-initialized CMIP6 ensemble is less pronounced in the tropical Pacific during summer, likely due to the peak activity of the El Niño Southern Oscillation (ENSO) in winter. Like climate predictions from dynamical forecasting systems, analog SPI3 forecasts are generally less skillful than 3-month TAS predictions but still show higher skill than the non-initialized CMIP6 ensemble and skill peaks around boreal winter. We show that skill in SPI3 predictions primarily stems from internal climate variability alignment,
while for TAS predictions, external forcing also plays an important role. Seasonal TAS and SPI3 predictions for all initializations throughout the year display clear added value over the non-initialized CMIP6 ensemble, but are generally less skillful than operational predictions from SEAS51 (Johnson et al., 2019). Over land these differences in skill between the analog-based and SEAS51 predictions are generally statistically not significant as measured by the MAESS. Furthermore, the spatial patterns of skill are very similar between the analog-based predictions and the state-of-the-art benchmark
prediction system SEAS51, suggesting that both predictions have skill due to similar physical processes.

On annual to multi-annual timescales, the annual TAS analog-based predictions are highly skillful across most tropical and many extratropical land regions. Central Asia, central North America, and southern South America show lower skill, but still better than climatological forecasts. The analog-based predictions generally outperform the CMIP6 ensemble, while the
added value over the CMIP6 ensemble decreases with increasing forecast range (i.e. biennial and quadrennial), indicating that external forcing drives most of the skill particularly at quadrennial timescales. Spatially, the skill of annual and biennial SPI forecasts is generally similar to that of seasonal ones, with positive statistically significant correlations in several tropical regions  being a common feature. High-latitude regions in Eurasia exhibit enhanced skill, particularly for quadrennial predictions, with external forcing contributing significantly to the skill in these areas. A comparison with the operational
decadal prediction system EC-Earth3  (Bilbao et al., 2021), reveals that the analog method can provide comparable annual and biennial predictions of TAS and SPI when the predictions are initialized at the same month (i.e. every November) and the lead-time increases. While decadal prediction systems are typically initialised only once per year, the analog-based predictions can however be easily generated every month in an operational context and the skill of those predictions is



broadly superior to the skill of the EC-Earth3 decadal predictions initialized only once a year. The 48-month analog-based
predictions of TAS and SPI are less skillful than the EC-Earth3 counterpart when initialized in November, but become
comparable if the analog-based predictions are produced every month.

Building on the established concept of climate analogs, our research demonstrates that by sampling through time and model
of a large CMIP6 multi-model ensemble based on their similarity with observed SST patterns, one can extract valuable
information on the future evolution of TAS and SPI, spanning a forecast range of seasons to multiple years. In other words,
the analog-based forecasts can provide seamless predictions for different forecast times, which have traditionally been
addressed with specific forecast systems (seasonal or decadal). These analog-based forecasts are competitive with the
existing prediction systems on annual to multi-annual forecast ranges. This methodology offers a complementary source of
climate information to existing seasonal and decadal climate predictions. Crucially, the method is computationally
inexpensive and based on a straightforward approach that facilitates the generation of seamless climate predictions
reproducible at low computational cost once the multi-model ensemble of transient simulations has been produced.

*Author contributions.* J.C.A.N., A.T. and A.H.E. conceived the idea for this study. J.C.A.N. conducted the analysis and
prepared all figures. P.D.L., M.D., R.M., D.V. and A.A. contributed with the preparation of CMIP6 and EC-Earth3 data. All
authors were actively involved in the analysis of the results and the writing process.

*Competing interests.* The authors declare no conflict of interest

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
