# Peer review of "Seamless seasonal to multi-annual predictions of temperature and standardized precipitation index by constraining transient climate model simulations"

_EGUsphere, 2025_

## Author Comment (AC1)

Comments Reviewer 1

MAJOR

L21: Given the lower skill of the analogs (e.g. Fig.1 and 2) but that they are potentially very useful as a tool for making seamless predictions, I think the abstract should make it clear that the skill is lower rather than 'competititve'.

This is a fair point, the text of the abstract (ln. 20-22) was changed to "The analog-based seamless prediction system  shows very similar patterns of skill compared to state-of-the art initialised climate prediction systems and has competitive skill- with initialized numerical prediction systems on annual and biennial forecast ranges. . "

L120: This presumably results in all members having the same trend? If so, this needs a little discussion in the text with pros and cons as you are losing the individual model response to forcing and replacing it with the multimodel mean. Does this also reduce the variance in the ensemble?

The trend adjustment is one of the necessary steps of the analog-based method due to the fact that analogs can be selected from any year in the period 1960-2030 and do not necessarily have the right forcing state. The trend adjustment ensures that if selecting analogs from a different year/forcing state this offset is corrected to actually represent the correct forcing of the year(s) of the predictions. To make this point clear we have modified sentences 132-136: "This is necessary because  the analogs can be selected from any year in the period 1960-2030 and do not necessarily have the right forcing state. The trend adjustment ensures that potential offsets related to selecting analogs from other forcing states are corrected to represent the forcing of the year(s) of the predictions"

L146: Also on trends. The reference forecast R is stated to be a trivial climatological forecast but what does this mean? Is it a constant climatological value for each variable? Why not use a linear trend for Ts? This would seem like a fairer test.

By 'climatological forecast,' we mean the climatological value for each variable in the specified period: December-February (Fig. 1-3), June-August (Fig. 2-3), and annual, biennial, and quadrennial climatology (Fig. 6, 7, and 8, respectively). We believe that besides the skill evaluation against a climatological forecast, which is a common practice in any skill evaluation of climate predictions, we have also presented the skill with respect to a more strict reference than a linear trend: the forced signal (i.e. Figs 1,2,3,6,7,8, panels b and e). The forced signal not only represents a trend that is non-linear, but also the signal stemming from external forcing of unpredictable events such as volcanic eruptions, hence making this a higher benchmark to compare to than a linear trend.

L160: Is it fair to compare ensembles of different sizes? There is plenty of literature on this point and all scores should either be calculated for the same ensemble size or corrected for ensemble size to make them equivalent. Even if large ensembles of analog forecasts are easy to generate, this is important for the comparison and understanding the relative merits of the methods.

We thank the reviewer for pointing this out. We agree that skill is sensitive to ensemble size (See Fig. S1 or Figures below). However, while the initialized seasonal and decadal prediction systems used as benchmarks require substantial computing cost for larger ensembles, the analog predictions can provide large ensemble sizes at no additional cost. Some skill advantage indeed results from the use of larger ensemble size. Just as the examples found in literature where a fair comparison is made between ensembles of different sizes, many others exist in which a larger multi-model ensemble is compared to the single system components (e.g. Hagedorn et al., 2005). Many times the advantage of multi-model ensembles is indeed related to their larger size. We have expanded a bit on this and clarified this in the manuscript in the lines 168-173: "Note that both dynamical prediction systems are limited to 25 members, whereas the analog-based predictions are based on the 149 members from the non-initialized CMIP6 ensemble. A key strength of the analog-based method is its ability to leverage a large-sized ensemble at minimal computational cost as opposed to the significant cost it requires to generate such large ensembles with initialized prediction systems. However, we acknowledge that a fraction of the skill of the analog-based predictions stems from exploiting large ensembles and reducing the ensemble size to match the size of the dynamical prediction systems reduces the skill. This is demonstrated in Fig. S1 which shows that the skill of the analog-based predictions clearly increases with ensemble size, regardless of variable or forecast range. (Fig. S1). ".

To further illustrate here the advantage of a large ensemble, the figures below show that indeed a 25-member analog-based prediction is generally less skillful than the 149 ensemble using the analog method, confirming the what Fig. S1 shows.

[Figure]

*Figure R1 (with 25-member analog ensemble): Fraction of land area with statistically significant positive correlation (p < 0.1) between the 3-month TAS (solid lines) and SPI3 (dashed lines) from the analog, a 25-member analog and SEAS51 predictions, and the respective observations. Statistical significance is assessed using a two-tailed t-test. The evaluation period is 1982-2018.*

[Figure]

*Figure R2 (with 25-member analog ensemble): Fraction of global land area with statistically significant (p<0.1) positive residual correlation between TAS predictions and observations for a) 12-month, b) 24-month and c) 48-month forecasts. Panels d), e) and f) are the same as a), b) and c), respectively, but for the statistically significant (p<0.1) positive correlation between SPI predictions and observations using a two-sided t-test. The dark green, purple and yellow lines in all panels show the skill of the analog-based, 25-member analog-based*

*and EC-Earth3 predictions, respectively, initialized every November as the lead-time increases from 2 to 13 months. The light green line shows the skill of the analog-based predictions, initialized always with a one-month lead-time.*

Fig. 4 has been modified from its original version to show the area with significant and positive skill for both TAS and SPI3, instead of the area with residual skill. We think that this comparison is more fair and since the skill does not saturate as it happens for longer forecast ranges, makes it a meaningful comparison. The conclusions remain qualitatively the same.

Note also that in response to the comments related to the trend removal, Fig. 11 has been slightly modified to account for the removal of trends using their own model trend. For example, EC-Earth3 residual skill is estimated by removing only the forced signal from EC-Earth3 uninitialized simulations. The conclusions also remain qualitatively the same.

Ref: Hagedorn, R., Doblas-Reyes, F. J., & Palmer, T. N. (2005). The rationale behind the success of multi-model ensembles in seasonal forecasting—I. Basic concept. Tellus A: Dynamic Meteorology and Oceanography, 57(3), 219-233.

L170-175, Fig2 and 3, L375: While I am sure readers will be open-minded to this method of forecasting this passage feels somewhat biased in favour of the analog method. The dynamical seasonal forecasts have a better correlation. This discussion needs to be rephrased and a panel of the difference in correlation scores is also needed, perhaps in place of the current panel 1b and panel 3b.

The figure below (Fig. R3) displays the difference in ACC between the analog-based predictions and SEAS51 for DJF and JJA forecasts of TAS and SPI3. Note that panel f in Figs. 1-2 already displays the direct comparison between the analog-based and the SEAS51 predictions with a dedicated discussion and it is comparable with the differences in ACC shown below. We therefore feel that showing the differences in correlation is not necessary. We also think that panels b in Figs. 1-3 are necessary as they show the skill of the forecasts after removing the forced signal. To better address the point of Reviewer 1 and avoid any bias in favor of the analog method we have removed the following sentence in L174-175 "*It is worth noting that skill over land of the analog-based predictions is in general statistically not different (p<0.1) between the analog-based and SEAS51 predictions as quantified by MAESS.*", and additionally changed sentence L375 as: " The analog-based predictions provide skilful forecasts on the seasonal to multi-annual time scales and show in general similar spatial patterns of skill to initialized numerical predictions. Furthermore, the analog-based predictions are competitive with existing annual and multi-annual predictions from initialized numerical predictions."

[Figure]

*Figure R3: Difference in TAS skill (anomaly correlation coefficients) between analog-based ensemble mean predictions and SEAS51 in a) December-February and b) June-August. Panels c) and d), show the same as a) and b), respectively, but for SPI3.*

Fig.4 and Fig.11: I think it is important that these metrics are changed to the *average correlation skill over land where it is significant*, rather than just the *area that is significant* because the current metric does not reflect the higher skill of SEAS5 in many regions and this is important for the value of the forecasts.

We disagree with the suggested method for displaying a summary of skill. Showing the *average of correlation coefficients only where it is significant* could yield biased results, but more importantly averaging correlation coefficients is not mathematically valid. For example, having a prediction with very little skill and only a few locations with high, but statistically significant correlation would give a high skill metric, whereas a prediction with widespread statistically significant correlation, but with low values would give a much lower metric, suggesting that is worse than the former prediction. In any case, we have recomputed skill in Figs. 4 and 11 (Figs. R4-R5 below) with the method suggested by the reviewer and see that both methods for computing the average skill yield qualitatively similar results. For these reasons we have decided to keep the original Figs. 4 and 11.

a

[Figure]

*Figure R4: Spatial average of skill over land that is positive and statistically significant (p<0.1), measured with the correlation of the 3-month TAS (solid lines), the SPI3 (dashed lines) from the analogs, and SEAS51 predictions, and the respective observations using a two-tailed t-test. The evaluation period is 1982-2018. The displayed values are based on residual correlations for TAS to remove the impact from the forced signal.*

[Figure]

*Figure R5: Spatial average of skill over land that is positive and statistically significant (p<0.1), measured with the residual correlation between TAS predictions and observations for a) 12-month, b) 24-month and c) 48-month forecasts. Panels d), e) and f) are the same as a), b) and c), respectively, but for the statistically significant (p<0.1) positive correlation between SPI predictions and observations using a two-sided t-test. The dark green and yellow lines in all panels show the skill of the analog-based and EC-Earth3 predictions,*

*respectively, initialized every November as the lead-time increases from 2 to 13 months. The light green line shows the skill of the analog-based predictions, initialized always with a one month lead-time.*

L250: in fact all the indices are of weak amplitude (even Nino3.4) so this needs to be stated with some comments about the ability to recalibrate the amplitude.

This is a good point, we have included the following sentence in L250 to acknowledge this point: "This implies that a recalibration of the ensemble could render the analog-based forecasts more valuable."

Fig.6: The analogs are clearly more competitive on this longer timescale and the striking similarity with the dynamical model is impressive, at least with EC-EARTH. However, I am not convinced EC EARTH is the best decadal prediction system. Does this result hold for other models? Either way, I think the abstract should reflect the benefit of analogs may be greater for the longer timescales.

The scope of the study was to show that the method has a comparable skill to an operational forecasting system. Whether or not EC-Earth is the best decadal forecast system is arguable, but from the skill analysis presented by the WMO lead center on decadal predictions (https://hadleyserver.metoffice.gov.uk/wmolc/), EC-Earth (BSC and SMHI/DMI) is ranked in the top half of models for both temperature and precipitation and forecasts of 1 and 5 years (Figures R6-R7 below). Additionally, the analog method is better than any single system for TAS forecasts of year 1 and years 1-5, and almost as high as the multi-system ensemble from all the producing centers. For precipitation (SPI12 and SPI60) the analog method is better than most single systems, but not as high as the multi-system ensemble (See figures below). We added the following sentence in lines in the summary and conclusions section "We have chosen EC-Earth3 as representative model of the typical decadal prediction system. It is possible that other decadal prediction systems perform better in particular regions and timescales, but EC-Earth3 forecasts quality metrics reveal it to be a good representative of these systems. For a thorough evaluation of several decadal prediction systems including EC-Earth, the reader is referred to Figures S7-S11 in Delgado-Torres et al., (2022). "

New reference:

Delgado-Torres, C., Donat, M. G., Gonzalez-Reviriego, N., Caron, L., Athanasiadis, P. J., Bretonnière, P., Dunstone, N. J., Ho, A., Nicoli, D., Pankatz, K., Paxian, A., Pérez-Zanón, N., Cabré, M. S., Solaraju-Murali, B., Soret, A., & Doblas-Reyes, F. J.: Multi-Model Forecast Quality Assessment of CMIP6 Decadal Predictions. *Journal of Climate*, *35*(13), 4363-4382, https://doi.org/10.1175/JCLI-D-21-0811.1, 2022.

[Figure]

Figure R6: Anomaly correlation between different decadal prediction systems and observations contributing to the WMO decadal predictions for 12-month (left) and 60-month TAS predictions (right). The decadal multi-model and the 149-member analog-based skill maps are shown on the top left and right, respectively.

[Figure]

Figure R7: The same as Fig. R5, but for precipitation (decadal prediction systems) and SPI12 and 60, for the analog-based predictions.

Fig.8e: Presumably this result comes from the fact that the analogs can be selected from any year? Does it improve if the analogues have to be selected e.g. from the same decade as the target? Or is this already accounted for by the removal and replacement of the forced trend?

Selecting the analogs from the same decade as the target would likely improve the trend but it also would reduce reduce the pool of analogs to select from, for example if the observations show a strong positive ENSO state, having only 10 years in the model would largely limit the available states and very possibly the part of the skill that is not externally forced. The figure below shows the metric for analog selection (left) and the years of selection of the analogs for each member based on that metric for an example prediction around the center of the period (year 2000) (See Fig R8). It is indeed accounted for by the removal and replacement of the forced trend (See answer to 2nd point above).

[Figure]

Figure R8: Example of the construction of an analog forecast for year 2000. Left) Globally-averaged mean absolute error of monthly model SSTs (vs. observations) as a function of ensemble member (x-axis) and model year (y-axis). Right) Red dots indicate the

*year of the 5 analogs selected for each member used to construct the analogs. Lowest values in purple and highest values in red.*

MINOR

L52: 'is meant to constitute a pool...' of course it does not always achieve this

Thank you, suggestion added.

L55: the number is not very small as it is now over 10 on subseasonal, seasonal and decadal scales. See for example Kumar et al, 2024, BAMS. Suggest to say "limited number"

The sentence L55 was changed to: "thus being produced only by a limited number of institutions around the world."

L64: '...of a more sophisticated'

Suggestion added.

L64: it is stated earlier that models drift to their own climatology and that this reduces skill. However L64 states that the analog method is not subject to drift because the model is in its own climate. This seems very one sided in favour of the analog approach and so it needs to be rephrased.

In reality the initialization shock is the largest concern, for that reason the word drift has been deleted from the sentence.

L70 Kushnir et al., 2019, Nat. C.C. is an important missing reference on the operationalisation of decadal predictions.

Included.

L80-85: please state the total sample size (in years), is it really greater than the decadal hindcast size?

Added. 10579 years in total.

L104: constraint

Changed.

L215: there is a long literature on Sahel forecasts so please add some references here.

References added.

Fig.10: please reduce the vertical scale to better show the variability.

Figure 10 has been changed accordingly.

L340: Smith et al 2018 specifically examined the ability of GCMs to predict global temperature: Smith et al, 2018. Predicted chance that global warming will temporarily exceed 1.5C. Geophys. Res. Lett.

We don't see how this reference fits well in line 340, for this reason it has been omitted.

---

## Author Comment (AC2)

Comments Reviewer 2

In the manuscript entitled "*Seamless seasonal to multi-annual predictions of temperature and standardized precipitation index by constraining transient climate model simulations*", Acosta Navarro and colleagues used an analog method to provide seamless climate forecasts of temperature and SPI across seasonal to multiannual timescales. This method has the advantage of providing forecasts that are not impacted by initialization shocks or drift and that can easily be updated monthly. I found that the method proposed by the authors, as well as the evaluation carried out, is interesting and will be of interest to the readers of the journal. However, I have some major and minor issues and comments that I hope are constructive, especially regarding the way the skill of the analog method is presented, as well as some methodological aspects.

Abstract:

l.17-18: Although the analog method generally provides better skill than the unconstrained CMIP6 ensemble mean, this is not always the case, with some regions consistently showing non-significant improvements. For example, this applies to large parts of the Northern Hemisphere continent in the seasonal prediction of surface temperature (Figs. 1 and 2e). For multiannual prediction, we can clearly see some regions where the analog method is less skillful than the unconstrained CMIP6 ensemble (Figs. 7 and 8e). Therefore, I believe the statement 'consistently outperforming the unconstrained CMIP6 ensemble' is somewhat biased and should be more nuanced.

This is a fair point, the text of the abstract was changed to "The analog method yields predictive skill for surface air temperature forecasts across timescales, ranging from seasons to several years., On average, the analog-based surface air temperature predictions provide added value over  the unconstrained CMIP6 ensemble, especially on seasonal to annual timescales."

l.20-22: The skill of the analog method is generally lower than that of seasonal prediction (e.g., Fig. 4). For multiannual prediction, the results are more mixed: the analog method appears to be slightly more effective than the EC-Earth3 prediction system for 12- and 24-month predictions but clearly shows lower skill at 48-month predictions (e.g., Fig. 11). Therefore, I believe the statement 'competitive compared to state-of-the-art initialized climate prediction systems' should again be more nuanced, emphasizing the method's potential at annual to biennial timescales, where it seems more competitive with state-of-the-art initialized climate prediction systems. Additionally, this does not detract from the fact that this method could be a highly useful tool for seamless predictions.

Reviewer 1 had a very similar comment, and we use the same response here:

This is a fair point, the text of the abstract (ln. 20-22) was changed to "The analog-based seamless prediction system  shows very similar patterns of skill compared to state-of-the art initialised climate prediction systems and has competitive skill- with

initialized numerical prediction systems on annual and biennial forecast ranges. . ”

Method:

l.93: How was the 1960–2030 period chosen? What is the added value of selecting analogs over a near-future period (i.e., selecting analogs from 2030 onward for a 2024 forecast, if I understand the method correctly)?

The period was chosen to be representative of the hindcast period 1962-2018, but also with the aim of keeping the future period in the method for performing actual forecasts. We think it is important that the near-future period is included in case unprecedented conditions happen in an operational context. For example the year 2024 was exceptional and most likely better represented by future CMIP6 conditions. The following sentence was added to clarify this point: "The period 1960-2030 was chosen to include a climatically representative period of the hindcast with an extension to the future to allow for the occurrence of unprecedented climatic states in a real-time forecasting context."

Table S4-5: Why use 4-month tests instead of 3-month tests, as in the other table, for the 24-month prediction?

This was a typo and we did it for 3 months instead of 4 to be consistent with 3- and 24-month predictions. it was corrected in the tables S4-S5.

l.105-106: Why the period used is smaller for longer prediction ?

This is simply limited by the data available between 1960-2030. For 4 year predictions, 4 year averages need to be made limiting the actual period on both ends. This is explained in the methodology section with an example in line 111.

l.114: I am a bit confused about Method 3. I understand that the authors aim to maintain a similar ensemble size while maximizing the number of models, but I am unsure how to interpret the results, especially in cases where the same member is chosen for the majority of the five analogs.

This was indeed the case only for models that had less than 10 members available. In those cases the same analog was repeated to guarantee an identical ensemble size for each model. This essentially means that the weight of those ensembles of models were increased. In any case the results seem very insensitive to the method chosen, yielding very similar overall results regardless of the method selected. We have modified the l. 116 to expand and clarify better this point:

Method 3: Ten members from each model. For models that provide fewer than 10 members, the members are used more than once to complete a set of 10 for each model, 190 members in total. Essentially increasing the relative weight of the analogs of models with fewer than 10 members.

l.119-123: The method used to remove the trend may need further clarification. Are you referring to removing a linear trend from the analog-method predictions and observations (this is not specified), as indicated in Figure 5 of Smith et al. (2019) ? If so, the potential implications should be clarified, as this approach may not effectively remove the forced signal compared to using the ensemble mean for each model. Additionally, the implications of this methodological choice—where all members will have the same trend—should be discussed further.

Same reply used in one of the comments from Reviewer 1:

We only performed a skill assessment of the ensemble mean of the analog-based predictions. For that reason removing the forced signal (CMIP6 ensemble mean) from the ensemble mean of the analog predictions is valid. The trend adjustment is one of the necessary steps of the analog-based method due to the fact that analogs can be selected from any year in the period 1960-2030 and do not necessarily have the right forcing state. The trend adjustment guarantees that this erroneous forcing is corrected to actually represent the correct forcing of the year(s) of the predictions. The analog-method before applying the trend correction was not designed to capture the forced response. To make this point clear we have modified sentences 121-125: "This step is done because  the analogs can be selected from any year in the period 1960-2030 and do not necessarily have the right forcing state. The trend adjustment guarantees that this erroneous forcing is corrected to represent the forcing of the year(s) of the predictions"

121-122: Does this mean that the analog-based method failed to capture the forced response in surface temperature ?

Please see the previous response. Although the analog-method captures trends, it underestimates them due to the nature of the methodology that selects analogs from different forcing states.  The trend adjustment is a necessary step in the generation of TAS predictions.

l.137-138: I am a bit confused here. For surface temperature predictions, you removed the external forcing, added the CMIP6 ensemble mean, and then used linear regression at each grid point to remove the CMIP6 ensemble mean in order to estimate internal variability. Why not directly estimate internal variability as the residual after removing the external forcing, before adding the CMIP6 ensemble mean?

It seems like there is likely a misunderstanding, we actually did exactly as the reviewer suggests. We have modified the text as follows: "Using the best overall method from the sensitivity tests (method 4), the selected analogs then constitute the forecasts and can be interpreted as ensemble members. Additionally, the trend  the ensemble-mean TAS

analog-based predictions is adjusted by first removing the signal explained by external forcing as in Smith et al., (2019) and then adding  the externally forced trend (i.e. the CMIP6 ensemble mean) to those residuals. To make this point clear we have modified sentences 132-136: "This is necessary because  the analogs can be selected from any year in the period 1960-2030 and do not necessarily have the right forcing state. The trend adjustment ensures that potential offsets related to selecting analogs from other forcing states are corrected to represent the forcing of the year(s) of the predictions. "

We additionally deleted the sentence in the paragraph below causing confusion: ""

l158: Why did you choose the EC-Earth3 prediction system, given that several centers now provide such predictions? It would be interesting to see whether the regions where the analog method performs better or worse than the EC-Earth3 prediction system remain the same for another prediction system. Additionally, I'm curious whether there is any known bias in the EC-Earth3 predictions that the analog-based method might improve.

Reviewer 1 had a very similar comment, and we include here part of that earlier response::

The scope of the study was to show that the method has a comparable skill to an operational forecasting system. We added the following sentence in lines in the summary and conclusions section and the figure below shows skill maps for TAS and precipitation for 12 and 60 month predictions. From the maps below (Figs. R6-R7) and the study from Delgado-Torres et al., (2022), it is clear that the skill patterns of the analog methods and the prediction systems share many similarities, including EC-Earth3. Looking into the particularities of a reference system was beyond the scope of the paper. The following lines were added in the summary and conclusions' section: "We have chosen EC-Earth3 as representative model of the typical decadal prediction system. It is possible that other decadal prediction systems perform better in particular regions and timescales, but EC-Earth3 forecasts quality metrics reveal it to be a good representative of these systems. For a thorough evaluation of several decadal prediction systems including EC-Earth, the reader is referred to Delgado-Torres et al., (2022). "

New reference:

Delgado-Torres, C., Donat, M. G., Gonzalez-Reviriego, N., Caron, L., Athanasiadis, P. J., Bretonnière, P., Dunstone, N. J., Ho, A., Nicoli, D., Pankatz, K., Paxian, A., Pérez-Zanón, N., Cabré, M. S., Solaraju-Murali, B., Soret, A., & Doblas-Reyes, F. J.: Multi-Model Forecast Quality Assessment of CMIP6 Decadal Predictions. *Journal of Climate, 35*(13), 4363-4382, https://doi.org/10.1175/JCLI-D-21-0811.1, 2022.

[Figure]

Figure R6: Anomaly correlation between different decadal prediction systems and observations contributing to the WMO decadal predictions for 12-month (left) and 60-month TAS predictions (right). The decadal multi-model and the 149-memeber analog-based skill maps are shown on the top left and right, respectively.

[Figure]

Figure R7: The same as Fig. R5, but for precipitation (decadal prediction systems) and SPI12 and 60, for the analog-based predictions.

Results:

l.175-178: Although the spatial pattern between the analog method and the prediction system is quite similar, the prediction system seems to have an overall larger correlation. A map of the difference between both would help clarify this, perhaps instead of Fig. 1c? This also seems to be the case for Fig. 2.

Same reply used in one of the comments from Reviewer 1:

The figure below (Fig. R2) displays the difference in ACC between the analog-based predictions and SEAS51 for DJF and JJA forecasts of TAS and SPI3. Note that panel f in Figs. 1-2 already displays the direct comparison between the analog-based and the SEAS51 predictions with a dedicated discussion and it is comparable with the differences in ACC shown below. We therefore feel that showing the differences in correlation are not necessary as we think keeping the panel c in these figures is important, because it shows the similarity of the skill patterns.

[Figure]

Figure R2: Difference in TAS skill (anomaly correlation coefficients) between analog-based ensemble mean predictions and SEAS5 in a) December-February and b) June-August. Panels c) and d), show the same as a) and b), respectively, but for SPI3.

l.215: "in which the skill seems to result from the external forcing." It would be nice to add some reference to support this point.

The following reference was added to the manuscript.

Ndiaye, C. D., Mohino, E., Mignot, J., & Sall, S. M.: On the Detection of Externally Forced Decadal Modulations of the Sahel Rainfall over the Whole Twentieth Century in the CMIP6 Ensemble, Journal of Climate, 35(21), 6939-6954, https://doi.org/10.1175/JCLI-D-21-0585.1, 2022

l.230: I would not say that the skill is 'comparable' for TAS, as there are some seasons for which the SEAS51 predictions have a global land fraction significantly more correlated with the observations, with values more than 10% higher than those for the analog method.

Thank you for raising this fair point. The whole section has been changed due to concerns from both reviewers related to this particular point. The passage now reads: "The  SPI3 analog-based predictions show skill  comparable to SEAS51 predictions  between boreal fall  and spring  in terms of land area with positive and statistically significant residual correlation , while the analog-based predictions of 3-month TAS are generally less skillful than SEAS51 throughout the year. Skill over land peaks around boreal summer/fall and fall/winter for TAS and SPI3, respectively in both analog and SEAS51 predictions. This difference between the two variables can most likely be attributed to a more dominant influence of external forcing on TAS predictability, while for SPI3 the primary driver is natural variability. "

l.271: Are you talking about model bias that influence the analog-method or bias in the analog-method results ? If it is the first one, references would be welcome here.

Biases in the analog method, now it is specified in the text.

l.272: As for Fig 1 and 2, it would be nice to see the map of the difference.

Same reply used in one of the comments from Reviewer 1:

The figure below (Fig. R1) displays the difference in ACC between the analog-based predictions and SEAS51 for DJF and JJA forecasts of TAS and SPI3. Note that panel f in Figs. 1-2 already displays the direct comparison between the analog-based and the SEAS51 predictions with a dedicated discussion and it is comparable with the differences in ACC shown below. We therefore feel that showing the differences in correlation are not necessary as we think keeping the panel c in these figures is important, because it shows the similarity of the skill patterns.

[Figure]

*Figure R1: Difference in TAS skill (anomaly correlation coefficients) between analog-based ensemble mean predictions and SEAS5 in a) December-February and b) June-August. Panels c) and d), show the same as a) and b), respectively, but for SPI3.*

Fig 11: It is interesting to see that the analog-based method is very close to EC-Earth3, or even slightly better, in terms of the fraction of global land area that is statistically significant with the observations for 12- and 24-month predictions of residual temperature and SPI. However, for 48-month predictions, the EC-Earth3 prediction system appears to be better. Do you have any thoughts on why the analog-based method might perform worse than the EC-Earth3 prediction system for long-term predictions?

We think that perhaps a longer period of SSTs constraints that define the analogs can play an important role at longer times. In this study we calculated the analogs based on SST average anomalies over one month, which was found optimal for the seamless seasonal to inter-annual time scale of predictions. However, for longer multi-annual to decadal prediction time scales, SST averages over longer periods (e.g. a few years) were found to be optimal for the analog definition e.g. by Mahmood et al. 2022 and Donat et al. 2024. The methods already contain some discussion related to this as well as the sensitivity tests available in the Supplementary material. The sentence was modified as: "Additional sensitivity tests also reveal that the optimal length (m and k) of SST pattern comparison is one month, independent of the different forecast ranges considered,  for seasonal to inter-annual predictions (Tables S2-S7).  Please note that for longer (e.g. multi-annual to multi-decadal) forecast times analogs based on longer-term SST averages were determined to give highest skill (e.g. Mahmood et al. 2022, Donat et al. 2024). The

time scales of the analogs represent processes relevant for the predictions. While for seasonal to inter-annual predictions SST variations at higher frequency (e.g. ENSO) are most relevant, for longer prediction horizons (also reaching beyond the ENSO predictability barrier) other lower-frequency variations (e.g. Atlantic Multidecadal Variability) are more relevant. "

l.366-367: I think this needs a bit more clarification here or in the Fig. 11 legend to make it easier for the reader to follow the analysis. In Fig. 11, if I understand correctly, the months correspond to the predictions for each month relative to the forecast time. For example, June for TAS 12-months corresponds to the temperature prediction for the first month of June in the forecast, and June for TAS 24-months corresponds to the temperature prediction for the second month of June in the forecast, with the prediction starting in November. Does the light green line represent the same thing, but starting in May instead of November? In that case, we can expect this result, as the prediction time is shorter for the light green curve.

The forecast ranges evaluated are always 12, 24 and 48 months for panels a,d, b,e and c,f, respectively. The month in the x-axis indicates the first month of each one of those periods of evaluation. For example, in Fig. 11a, the value of March, means that the predictions were evaluated between March and the following February. The lead time increases for EC-Earth3 and analogs, since they are initialized each November, while the analogs-1 month lead is always initialized the month before the first month of evaluation, in this example, each February.

The following sentence was added in the end of the caption of the Figure: The x-axis always shows the first month of the forecasted period evaluated. For example, in panels b,e the values of August indicate the skill for predictions between August in the first forecast year and July two years later.

Note also that due to the previous comments related to the trend removal, Fig. 11 has been slightly modified (See 4th comment from Reviewer 1) to account for the removal of trends using their own model trend. For example, EC-Earth3 residual skill is estimated by removing only the forced signal from EC-Earth3 uninitialized simulations, while the 25-member analog-based predictions use the same 25 members from the uninitialized ensemble to compute the residual skill. The conclusions remain qualitatively the same.

l.375-376: Same comments as for the abstract.

These sentences have been changed due to concerns from both reviewers, now it reads: " The analog-based predictions provide skilful forecasts on seasonal to multi-annual time scales and show in general similar spatial patterns of skill to initialized numerical predictions. Furthermore, the analog-based

predictions are competitive with existing annual and multi-annual predictions from initialized numerical predictions. "

I.403-405: This is a strong added value of the analog method, I think it should be emphasized more

Sentence in L444 has been modified to highlight more this advantage: "This methodology offers a complementary source of climate information to existing seasonal and decadal climate predictions, filling an existing gap across timescales and doing so in a seamless manner."

I.408-416: I would just add a point to remind that the analog-method does not induce any drift due to the shock of the initialization, which is also an important added value of the method.

The sentence in line 441 has been changed to mention this: "Despite some potential limitations related to the lack of a more sophisticated model initialization, These analog-based forecasts have no initialization shock nor drift, and are competitive with the existing prediction systems on annual to multi-annual forecast ranges. "

Small correction:

I.99: The reference is not complete

Corrected.

l265-266: "The residual correlation of the analog forecasts is illustrated in Figure 6b." → This information is already in the legend, so I'm not sure how useful it is here.

Good point, the sentence was deleted.

l302: I think it is Fig 8 instead of Fig 9

It has been corrected.

---

## Referee Report (RR1)

**Review of "Seamless seasonal to multi-annual predictions of temperature and standardized precipitation index by constraining transient climate model simulations"**

I like the method presented here as producing (cheap, and therefore more flexible) forecasts that are complimentary to the traditional seasonal to decadal forecasts using an initialised climate model. I don't have many other comments and I think the initial reviewer comments have been well addressed by the authors.

**Small typos:**

L 37: "associated with phenomena"

L 64: "lack of a more"

L 65: Remove the full stop between "shocks" and "and"

L 298: "Africa and Australia"

L 417: as a representative model

L 431: seamless manner. Crucially" (add the missing full stop)

**Main comments:**

1. "A key strength of the analog-based method is its ability to leverage a large-sized ensemble at minimal computational cost as opposed to the significant cost it requires to generate such large ensembles with initialized prediction systems." (L 169)

I mostly agree with this statement, however is there a slight nuance here? The method you use for analog selection is a simple ranking of the MAE, so as you increase the analog ensemble size, are you also degrading the ensemble selection by selecting ensemble members that are further away from the initial state? You demonstrate in the comments to the original reviewer that 149 members are generally more skilful that 25, but at some point, that will not be true.

- 2. The discussion about Figure 4 around line 240: Could a reference to the figure number be added in the text?
- 3. "This implies that a recalibration of the ensemble could render the analog-based forecasts 260 more valuable." (L 259)

Could you add to this comment? I'm not sure how the recalibration could be done. Would the recalibration be done to the field before choosing the analogs? Could it be that the reduction in variability is due to the analog sampling. If many approximate matches are sampled and average, the size of the signal may be reduced?

---

## Author Response (AR2)

**Comments Reviewer 1**

I like the method presented here as producing (cheap, and therefore more flexible) forecasts that are complimentary to the traditional seasonal to decadal forecasts using an initialised climate model. I don't have many other comments and I think the initial reviewer comments have been well addressed by the authors.

We thank the reviewer for taking the time to review the manuscript once again and for providing constructive comments.

Small typos:

L 37: "associated with phenomena"

It has been changed.

L 64: "lack of a more"

It has been changed.

L 65: Remove the full stop between "shocks" and "and"

It has been changed.

L 298: "Africa and Australia"

It has been changed.

L 417: as a representative model

It has been changed.

L 431: seamless manner. Crucially" (add the missing full stop)

It has been changed.

Main comments:

1. "A key strength of the analog-based method is its ability to leverage a large-sized ensemble at minimal computational cost as opposed to the significant cost it requires to generate such large ensembles with initialized prediction systems." (L 169)

I mostly agree with this statement, however is there a slight nuance here? The method you use for analog selection is a simple ranking of the MAE, so as you increase the analog ensemble size, are you also degrading the ensemble selection by selecting ensemble members that are further away from the initial state? You demonstrate in the comments to the original reviewer that 149 members are generally more skilful that 25, but at some point, that will not be true.

Figure S1 in the supplementary material shows the fraction of land area with statistically significant positive correlation as a function of ensemble size for all variables studied (SPI-

- 3, 12, 24 and 48 and TAS 3, 12, 24 and 48 months). It is quite clear from the figure that the skill of the forecasts increases with ensemble size with no sign of reaching a maxima with ensembles smaller than 149, which is our ensemble size maximum. We agree with the reviewer about the potential shortcomings of selecting large ensembles using the analog method. In the previous review we had written in the methods section (L62-65) the following sentences to make it clear "An incomplete representation of the climate state at initialization is likely the major disadvantage of the analog-based predictions because of the finite available states present in the multi-model catalog. These states may be less representative or "further away" from the observed target state than the initial states in an initialized climate prediction system."
- 2. The discussion about Figure 4 around line 240: Could a reference to the figure number be added in the text?

**It has been added.**

3. "This implies that a recalibration of the ensemble could render the analog-based forecasts 260 more valuable." (L 259)

Could you add to this comment? I'm not sure how the recalibration could be done. Would the recalibration be done to the field before choosing the analogs? Could it be that the reduction in variability is due to the analog sampling. If many approximate matches are sampled and average, the size of the signal may be reduced?

This is a good point. We think that the only possible calibration would have to be done a posteriori as it is usually done for other predictions. We have added before line 239 the following sentences to address the comment. "This underestimation of the local variability may be a result of the sampling and averaging of hundreds of analogs with some of them being less representative of the observed conditions in the particular area. Despite this, at the global scale, the larger the ensemble of analogs the higher the skill (Fig. S1). As commonly done for other types of climate predictions, recalibrating the ensemble could make the analog-based forecasts more valuable."